



# New insights into radiative transfer in sea ice derived from autonomous ice internal measurements

Christian Katlein[1,2], Lovro Valcic[3], Simon Lambert-Girard[2], Mario Hoppmann[1]

5 [1] Alfred-Wegener-Institut Hemholtz-Zentrum für Polar- und Meeresforschung, Sea Ice Physics, Bremerhaven, Germany
[2] Takuvik Joint International Laboratory, Université Laval and CNRS (France), Québec, QC, Canada
[3] Bruncin Observation Systems, Zagreb, Croatia

10 *Correspondence to*: Christian Katlein (ckatlein@awi.de)

Submission to: The Cryosphere

15 **Abstract.** The radiative transfer of short-wave solar radiation through the sea ice cover of the polar oceans is a crucial aspect of energy partitioning at the atmosphere-ice-ocean interface. A detailed understanding of how sunlight is reflected and transmitted by the sea ice cover is needed for an accurate representation of critical processes in climate and ecosystem models, such as the ice-albedo feedback. Due to the challenges associated with ice internal measurements, most information about radiative transfer in sea ice has been gained by optical measurements above and below the sea ice. To improve our 20 understanding of radiative transfer processes within the ice itself, we developed a new kind of instrument equipped with a number of multispectral light sensors that can be frozen into the ice. A first prototype consisting of a 2.3m long chain of 48 sideward planar irradiance sensors with a vertical spacing of 0.05 m was deployed at the geographic North Pole in late August 2018, providing autonomous, vertically resolved light measurements within the ice cover during the autumn season. Here we present the first results of this instrument, discuss the advantages and application of the prototype and provide first 25 new insights into the spatiotemporal aspect of radiative transfer within the sea ice itself. In particular, we investigate how measured attenuation coefficients relate to the optical properties of the ice pack, and show that sideward planar irradiance measurements are equivalent to measurements of total scalar irradiance.

## 1 Introduction

The optical properties of the sea ice covering the polar oceans are a crucial parameter in the Earth's climate system [*Grenfell 30 et al.*, 2006; *Perovich et al.*, 2007]. Reflection and transmission of sunlight by sea ice determine the partitioning of shortwave radiative energy, in particular the reflection of incident irradiance back into the atmosphere [*Curry et al.*, 1995; *Perovich*, 1990]. Light transmitted through the ice cover does not only heat the underlying ocean [*Steele et al.*, 2010], but also provides energy for the ice-associated ecosystem [*Assmy et al.*, 2017; *Leu et al.*, 2010]. With thinner and younger ice



[*Haas et al.*, 2008; *Renner et al.*, 2014] covering a smaller part of the ocean due to anthropogenic climate change [*Serreze et al.*, 2007; *Stroeve et al.*, 2012], the relative importance of the shortwave energy budget is increasing.

Optical properties of sea ice have been most frequently determined using measurements above and below the ice [*Eicken and Salganek*, 2010; *Grenfell et al.*, 2006; *Grenfell and Maykut*, 1977; *Perovich et al.*, 1998]. Despite recent efforts to increase the number of such measurements by the means of robotic platforms [*Katlein et al.*, 2019; *Katlein et al.*, 2015; *Katlein et al.*, 2017; *Nicolaus and Katlein*, 2013], these methods only provide bulk properties that do not resolve vertical variations of sea-ice optical properties. Optical properties within the ice have so far mostly been determined by the use of inverse radiative transfer models which fit a vertical profile of optical properties to observations above and below the ice [*Ehn and Mundy*, 2013; *Ehn et al.*, 2008b; *Light et al.*, 2008]. Furthermore, optical properties have been derived using various techniques of analyzing extracted ice samples in laboratory setups [*Grenfell and Hedrick*, 1983; *Katlein et al.*, 2014b; *Light et al.*, 2015]. Extraction of samples however is not only destructive to the natural sea ice environment, but samples also often undergo quite dramatic physical changes such as brine drainage or freezing between sample extraction and analysis in the lab.

[*Ehn and Mundy*, 2013; *Ehn et al.*, 2008a; *Light et al.*, 2008; *Pegau and Zaneveld*, 2000; *Xu et al.*, 2012] used vertically profiling light sensors to extract radiance or irradiance profiles and attenuation coefficients within holes drilled into the ice. [*Maffione et al.*, 1998; *Voss et al.*, 1992; *Zhao et al.*, 2010] used active measurements in drilled ice to measure the point spread function, beam spread function and horizontal light attenuation. These methods have in common that they have to be operated manually and thereby take significantly more resources compared to continuous measurements on automated platforms.

Autonomous continuous and vertically resolved optical measurements within the interior of sea ice have the potential to give a more comprehensive view on radiative transfer processes within the ice itself. They can provide closer bounds on the inherent optical properties of the ice interior, and - if combined for example with commonly available temperature chains – also provide new impulses towards the improvement of structural optical models. Light sensor chains are also the least invasive way to perform such measurements, while the small size also minimizes effects of self-shadowing. Moreover, recording data throughout the entire seasonal cycle allows for a better understanding of the seasonal evolution of the ice and how parameterizations for large scale models can be optimized. Multispectral optical data also allow for the detection of ice algae growing within the sea ice [*Katlein et al.*, 2014a; *Lange et al.*, 2016; *Mundy et al.*, 2007].

Here we present the concept, design and first results of a new autonomous in-ice light sensor chain that is able to provide multispectral in-ice light data at a significantly higher spatial and temporal resolution than previously feasible. Hourly measurements of multispectral in-ice sideward planar irradiance at a vertical resolution of 5 cm were acquired from a prototype system that was deployed in August 2018 close to the geographic North Pole. In addition to the description of the instrument itself, and a detailed analysis of this dataset, we also link the results to the physical background necessary for a proper interpretation of its data. Finally, we discuss advantages and limitations, and highlight its scientific potential towards future applications.



## 2    Materials and Methods

### 2.1    The chain design

To measure vertical irradiance profiles within the ice, we build on years of experience in the field of ice temperature
measurements using thermistor chains [*Hoppmann et al.*, 2015; *Jackson et al.*, 2013; *Perovich and Richter-Menge*, 2006;
*Planck et al.*, 2019; *Richter-Menge et al.*, 2006]. These chains provide an easily deployable tool for autonomous
measurements of the in-ice temperature field and the thermal properties of the ice column, out of which ice thickness and
snow depth can be derived. Only recently these systems have transitioned from traditional analog thermistor sensors to a
chain of digital temperature sensors mounted on flexible printed circuit boards similar to the design of LED lighting strips
[*Jackson et al.*, 2013].

For this light sensor chain design, we replaced the digital temperature sensors with multispectral light sensors protected by
transparent heat shrink tubing. This analogous design provides the same advantage of easy deployment through a 5cm (2")
diameter hole combined with a rather rugged form factor without protruding parts. The prototype chain presented here was
equipped with 48 sensors at 5cm spacing resulting in a total chain length of 2.4m. Longer chains, more sensors and different
sensor spacing can be implemented depending on needs. The chain can be deployed through any kind of ice. While
temperature sensor chains are crucially dependent on refreezing of the hole around the chain, the optical sensors are less
sensitive to a potential distance to the ice, as the effects of a small hole on the in-ice light field are compensated by the strong
multiple scattering in the ice and the large sideward viewing angle of the sensors as long as the top most dry part is refilled
with drill cuttings after the chain deployment.

### 85   2.2    The multispectral irradiance sensor

To achieve a low total system cost and to maintain the small and compact form factor, we make use of a sensor initially
designed for color and ambient light measurement in mobile and small devices. The only 2.0 by 2.4 mm small TCS3472
sensor (ams Sensors Germany GmbH, Jena, Germany) is a 3x4 photodiode array recording irradiance in four spectral bands
(Figure 1). This encompasses the typical red, green and blue (RGB) channels (Figure 2), as well as a "clear" channel
integrating all these wavelengths in the visible range. The spectral response of the clear channel closely resembles the ones
of commercially available photodiode PAR sensors. An infrared filter however limits the spectral sensitivity above 650nm in
comparison to high quality PAR sensors (Datasheet available at https://ams.com/tcs34725).

As the flat optical sensor is almost directly exposed, it exhibits a good cosine response and thus provides measurements of
planar irradiance. The photodiode signals are converted by analog-digital-converters on board the chip and transmitted to the
board controller using the I2C protocol. The chain consists of eight segments each containing eight sensors that are
controlled via a I2C interface chip. Thus the entire chain only relies on 4 continuous electric lines, two for data
communication to all sensors and two for 3.3V power supply. The sensor provides an output in uncalibrated counts that are
linearly proportional to the measured irradiance. The digital sensor provides excellent measurement stability and is rated for





use at temperatures down to -40°C. It offers a dynamic range of 3,800,000:1 and thus provides precise measurements both
under full illumination at the surface, as well as under 2m thick ice in the Arctic at downwelling planar irradiance fluxes
below 0.05 W/m². In the presented prototype unit, the sensor was always operated at a gain of 1x for highest data quality, but
higher gain values of 4x, 16x, 60x can be used to increase low-light sensitivity.

### 2.3    First deployment

After development and fabrication by Bruncin Observation Systems in spring 2018, the first prototype was deployed during
the AO18 expedition of the Swedish research icebreaker *Oden*. The icebreaker anchored for four weeks at an ice floe in
vicinity to the geographic North Pole. On 20 August 2018, the light chain system was deployed as part of a modular ice mass
balance and radiation station. Apart from the light chain, the system consisted of three RAMSES-ACC-VIS hyperspectral
radiometers (TriOS GmbH, Rastede, Germany) measuring downwelling, reflected and transmitted planar irradiance in the
wavelength range of 320-950 nm at 3nm spectral resolution. The RAMSES sensors and their associated data processing have
been described in detail by *Nicolaus et al.* [2010b]. In addition, it comprised a webcam, sensors for snow height, water
temperature, water salinity, as well as a thermistor chain measuring a vertical profile of ice temperature and thermal
conductivity. This installation was part of a setup of multiple autonomous measurement systems including a Snow Buoy
(MetOcean Telematics, Halifax, Canada), a bio-optical buoy, a IAOOS buoy with an atmospheric lidar and an ocean profiler
[*Gascard*, 2011], as well as a timelapse camera.

The light chain was deployed through a 5cm hole on bare ice of 2.05m thickness. The ice was homogenously grown, but
covered by a 10-15cm thick surface scattering layer. Backtracking by the IceTrack Algorithm [*Krumpen et al.*, 2019]
assigned an age of three years to this ice, which is in accordance to ice core salinity data (not shown). The light chain was
deployed approximately 1.5 m away from a melt pond, which might have influenced the measured in-ice radiation field
[*Petrich et al.*, 2012]. When the site was left on 15 September 2018, about 5cm of snow had fallen, reducing overall light
transmission.

Two weeks after deployment, on 3 September 2018, one of the control chips on the chain failed, probably due to physical
forces during refreezing or in-ice pressure. This failure disabled the 4[th] section of the chain and heavily influenced readings
from section 8. About 10 days later, a similar failure occurred in chain sections 2 and 6. All other chain sections provided
useful data, until absolute light levels dropped below 0.05 W/m² in the beginning of October. Total failure of the system
occurred in December 2018, when the entire system ceased data transmission for unknown reasons.

### 2.4    Calibration

For scientific use, the TCS3472 sensors along the chain have to be cross calibrated. Absolute calibration is generally not
necessary, since the data is mainly used as relative measurement between different sensors on the same chain. For different
applications, absolute calibration might be necessary. The calibration has to be performed separately for all four spectral
bands. To ensure a consistent calibration of the chain, it was strapped flat on the ship's railing with all sensors pointing





upwards for several days. To avoid effects of shadowing and different sensor views of the ship's superstructure, we only used data from strongly overcast weather conditions for the calibration. Each channel and all sensors were compared against the average along all chain sensors for each recorded time step. From this, individual calibration coefficients were derived, that were applied to each sensor and channel of the entire dataset before further processing. This procedure does not account
for calibration uncertainties in between channels. Retrieval of exact spectral ratios between channels would thus require a full absolute radiometric lab calibration of across all sensors and channels. Such a more sophisticated calibration could be performed by the manufacturer in a custom integration sphere beforehand of field-work but this would increase system cost dramatically.

## 2.5 Radiative transfer model

To evaluate the effect of the sideward-looking sensor geometry, we modeled the ice-internal radiance field with the radiative transfer model DORT 2002 version 3.0 [*Edström*, 2005]. DORT 2002 is an independent MATLAB implementation of the discrete ordinate radiative transfer model DISORT [*Hamre et al.*, 2004; *Laszlo et al.*, 2016; *Stamnes et al.*, 1988] specifically designed for easy application in highly scattering media. To approximate the ice geometry during the light chain deployment, we used a four layer model with the following typical inherent optical properties of multi-year ice: A
transparent atmosphere with fully isotropic downwelling radiance distribution; a 0.1 m thick surface scattering layer with an absorption coefficient $a = 0.15\ m^{-1}$, a scattering coefficient $b = 250 m^{-1}$ and a Henyey-Greenstein phase function with asymmetry parameter $g = 0.9$; a 2 m thick interior ice layer with an absorption coefficient $a = 0.15\ m^{-1}$, a scattering coefficient $b = 25 m^{-1}$ and a Henyey-Greenstein phase function ($g = 0.9$) as well as an underlying ocean with an absorption coefficient of $a = 0.15\ m^{-1}$ and a scattering coefficient $b = 0.1\ m^{-1}$. These parameters resulted in calculated ice
albedo and transmittance values very similar to observations.
Downwelling planar, downwelling scalar, as well as sideward planar irradiances were calculated from the resulting radiance distributions using Lebedev Quadrature [*Katlein et al.*, 2016; *Light et al.*, 2003].

## 3 Results

### 3.1 Vertical profile of sideward planar irradiance

Figure 4 shows the measured sensor response in the clear after along-chain cross-calibration. The sensor chain is successfully capturing the spatiotemporal variation of the in-ice light field from the surface through the ice to the underlying ocean. The dynamic range of the sensor is sufficient to cover both, the incoming light field as well as the under-ice light field. The expected decay of the light field with increasing depth becomes obvious in the measurements and, except for a few sensor failures, data could be recorded until solar incoming radiation significantly decreased by end of September 2018.
Figure 5 shows sample vertical profiles of sideward planar irradiance along the chain length, and in particular throughout the ice column. The strongest light attenuation is associated with the surface scattering layer and the top 0.2m of the ice. Below





0.3m, the decrease becomes more uniform until a depth of approximately 1.5m. The decrease is linear in the logarithmic plot, which means it closely follows an exponential decay law. This suggests that, in this interior part of the ice, the radiance distribution has reached the asymptotic limit. In this regime, the light attenuation is only dependent on the inherent optical properties of the medium, and not on local slab geometry or the incident light field. Also, the radiance distribution is constant with depth and thus all attenuation coefficients of radiance, downwelling planar irradiance, sideward planar irradiance and scalar irradiance are identical. Beneath 1.5m, the decay of sideward planar irradiance accelerates, representing increasing loss of photons into the underlying weakly scattering water. This apparent change in light attenuation is associated to the vicinity of an interface i.e. the underlying ocean and is not necessarily related to changes in the inherent optical properties of the ice. Under close investigation, the profiles in Figure 5 exhibit two layers with slightly different optical properties with stronger light attenuation between 0.3-1.0m than in the underlying layer between 1.0-1.5m. This is likely due to a different age and thus brine or bubble content of the two respective ice layers. Figure 5a also clearly shows that light in the red channel is attenuated significantly stronger than light in the other spectral channels, but otherwise results for the different channels are very similar.

## 3.2    Diffuse attenuation coefficients

Figure 6 shows the apparent diffuse attenuation coefficients of sideward planar irradiance as derived from neighboring sensor pairs. The vertical diffuse attenuation coefficients $\kappa_{i,j}$ of sideward planar irradiance were derived from pairs of neighboring values of sideward planar irradiance $(E_i, E_j)$ and the distance $d$ between sensors:

$$\kappa_{i,j} = -\frac{\ln\left(E_j/E_i\right)}{d}$$

Due to remaining calibration uncertainties, as well as the impact of macroscopic variations in the ice structure (e.g. large brine channels), the retrieved coefficients vary a lot between neighboring sensor pairs. However, this variation is consistent over time and thus could be accounted for by vertical smoothing.

The highest attenuation coefficients $> 20\ m^{-1}$ are associated with the air-ice interface, while they remain typically below $2\ m^{-1}$ for the ice interior. Towards the ice bottom, apparent attenuation coefficients increase to $4 - 10\ m^{-1}$ as the asymptotic regime is passed. This is caused by increasingly strong photon loss through the underside of the ice and not a change in the inherent optical properties of the ice. This layer moves upward with time, likely due to a reducing ice thickness as a result of bottom melt.

## 3.3    Temporal evolution of ice and snow optical properties

Figure 7 provides a close-up look on the temporal evolution of light attenuation in the first half meter of the chain. Comparing the attenuation data to snow height measurements by acoustic sensors deployed together with the chain as well as a Snow Buoy deployed in the vicinity of the light chain, shows clearly how an increasing snow cover increases the apparent light attenuation in the sensor pair at the air-ice (snow) interface. After a significant snow fall event on 14 September 2018,



the vertical position of the strongly attenuating layer shifted in accordance to the increased snow depth. Due to the strong scattering in the snow, radiative losses are highest directly in the uppermost centimeters, where apparent attenuation quickly reaches values above $20\ m^{-1}$ inside the snow layer. This high scattering of snow also causes that the radiance distribution

quickly reaches the asymptotic state, leading to a nearly exponential decay of light within the thin snow layer (Figure 5). The vertical sensor spacing of this particular light chain is 5cm, and therefore too coarse to determine snow optical properties precisely, and retrieved values of the uppermost snow layer are thus highly dependent on the actual geometric position of the interface between the two respective surface sensors. The uppermost sensors can also easily be influenced by frost, rime or snow deposited onto the chain by wind as well as local snow accumulations around the sensor chain that do not necessarily

represent the overall snow conditions. Such effects are likely causing the significant temporal variation in apparent attenuation coefficients of the top 5 cm e.g. between 30 August and 14 September.

Vertical and temporal profiles of attenuation coefficients for the ice interior are shown in Figure 8. Averaging the resulting apparent attenuation coefficient for each depth layer reveals a typical structure of light attenuation within sea ice. The topmost 0.15 m are characterized by very high attenuation in the so-called surface scattering layer, consisting of large

deteriorating ice grains (Figure 8a). From 0.4 to 1.8 m depth, the retrieved attenuation coefficients are representative of more homogenous interior sea ice. While local variations in optical properties in the direct vicinity of the sensor cause significant scatter between neighboring depth layers, several regimes of light attenuation are clearly discernable. An upper layer roughly between 0.3 and 1.0 m depth exhibits vertical attenuation coefficients around $1.5\ m^{-1}$, while further below values scatter around $0.5 - 1.0\ m^{-1}$. This likely corresponds to different annual growth layers with differences in optical properties of this

multiyear ice floe and is consistent with previous observations of light attenuation in interior ice [*Light et al.*, 2008]. Below 1.6 m, approximately 0.5 m away from the ice water interface, derived attenuation values start to increase. This is however not related to a vertical change in the optical properties, but to the increasing photon loss through the ice-water interface. As the scattering coefficient of clear Arctic seawater is much lower compared to ice, the number of photons scattered back to the ice by the water is reduced considerably.

When looking at a time series of vertically averaged (bulk) attenuation coefficients for the entire chain length (Figure 8b), we observe values increasing from $1.7\ m^{-1}$ to $2.1\ m^{-1}$ for the green, blue and clear channels. These values are consistent with typical bulk attenuation coefficients used in model parameterizations [*Grenfell and Maykut*, 1977; *Perovich*, 1996] and larger scale estimations of bulk attenuation coefficients during summer [*Katlein et al.*, 2019]. Bulk attenuation coefficients for the red channel are slightly higher, increasing from $2.2\ m^{-1}$ to $2.6\ m^{-1}$ as the absorption coefficient of ice and water is

higher in the red part of the spectrum [*Grenfell and Perovich*, 1981].

While the sensor spacing of 0.05 m is not sufficient to discriminate between different snow layers, it still enables us to also investigate the temporal evolution of optical properties in individual ice layers (Figure 8c,d). Of particular interest is here the ice interior between 0.3 and 1.8 m, where light attenuation is little affected by boundaries, and the light attenuation coefficient is directly related to the material-inherent optical properties. The retrieved values are in the same order as





previous observations, with attenuation coefficients for the clear channel rising from $0.8\ m^{-1}$ to $1.0\ m^{-1}$ and $1.1\ m^{-1}$ to

$1.3\ m^{-1}$ for the red channel [*Grenfell and Maykut*, 1977; *Light et al.*, 2008; *Perovich*, 1996].

### 3.4     Interpretation of sideward planar irradiance data

A main factor in the design of the presented light sensor chain is that sensors are oriented sideward in contrast to normal

radiation sensors that are usually oriented horizontally. Most autonomous radiation stations measure planar irradiance, as this

is an essential quantity for physical processes, describing the directional flux of light energy through a plane [*Nicolaus et al.*,

2010a; *Nicolaus et al.*, 2010b; *Wang et al.*, 2014]. For biological applications, scalar irradiance, which describe the non-

direction-depending total energy flux through a certain point in space, is however of greater importance [*Arrigo et al.*, 1991;

*Ehn et al.*, 2008a; *Morel and Smith*, 1974]. Radiative transfer modeling of the observed ice cover however allows us to

derive general relationships between sideward looking and horizontally oriented irradiance measurements. Figure 9a presents

profiles of downwelling planar, scalar and sideward planar irradiance in atmosphere, snow, ice and ocean as simulated using

the DORT2002 radiative transfer model (see section 2.5). It is evident that both, sideward planar and scalar irradiance,

exhibit stronger attenuation close to the ice bottom in comparison to planar downwelling irradiance. This is caused by

increasing photon loss through the ice-ocean interface. Photons travelling in the downwelling direction –which provide the

largest contribution to planar downwelling irradiance- are least affected by the proximity of the ice-ocean interface. Thus

attenuation of downwelling planar irradiance closely follows an exponential decrease with depth in a wider vertical range

compared to scalar and sideward planar irradiance.

The most important conclusion can be reached by comparing ratios of the three irradiance quantities (Figure 9b). The ratio of

sideward to planar irradiance is decreasing with depth: a ratio of 0.9 applies above sea ice. Values around 0.75 are

representative for the asymptotic regime in the ice interior, while a ratio $\leq 0.4$ is found in the underlying water column. For

the ratio of sideward to scalar irradiance, we see however that the ratio is close to 0.25 throughout all media. The measured

sideward irradiance data can thus easily be converted to total scalar irradiance by multiplication with a factor of 4. Any

measurement of sideward planar irradiance within a strongly scattering medium is thus essentially identical to a

measurement of total scalar irradiance. As scalar irradiance is particularly sought after for any studies of biological

productivity, this equivalency provides an efficient means to autonomously measure light levels within solid sea ice also

with high relevance for such studies. Direct sensing of downwelling planar irradiance would require horizontal sensors and

thus a larger geometric footprint and also result in a significant impact of self-shading.

Due to the multiple scattering nature of light transfer in sea ice, the diffuse attenuation coefficients can generally not be

directly inferred from the material's inherent optical properties. Comparing modeled attenuation coefficients of sideward

planar irradiance with two parameterizations from the field of ocean optics [*Kirk*, 1984; *Mobley*, 1994], which are not able to

reproduce the observed attenuation profile, clearly indicates this (Figure 9c). However it is evident that in the asymptotic

regime within the ice interior, the attenuation coefficients are closely linked to the material properties. Additionally, within

the asymptotic regime the attenuation coefficients for all three irradiance quantities are identical [*Mobley*, 1994], so that ice



internal attenuation coefficients for downwelling planar irradiance can be derived from the presented measurements of sideward planar irradiance. The asymptotic regime is characterized by the area far enough from medium boundaries, where

the angular radiance distribution is invariant with depth. This is quickly reached for the ice interior, in particular due to the overlaying, highly scattering snow and surface layers.

### 3.5    Comparison to classical hyperspectral setup

To assess the applicability and accuracy of the light sensor chain, we compare sea ice bulk transmittance derived from the chain with measurements from the co-deployed hyperspectral RAMSES radiometer station. Due to the horizontal orientation

of the surface sensor, such a derivation is highly sensitive to shadowing and azimuthal effects under clear sky, but can be very accurate for the highly isotropic radiance distribution frequently caused by the persistent low cloud cover in the summer Arctic.

Sea ice light transmittance $T$ in the respective spectral channels $(R, G, B, C)$ was derived by averaging sideward planar irradiance values $E$ of the first three sensors (# 1,2,3) along the chain above the air-ice (snow) interface and the last three

sensors (# 46, 47, 48) underneath the ice-ocean as $T_{R,G,B,C} = (\overline{E_{46}, E_{47}, E_{48}})/(\overline{E_1, E_2, E_3})$. RAMSES hyperspectral radiometer measurements above $(E_{in}(\lambda))$ and below $(E_{trans}(\lambda))$ the ice were folded with the spectral sensitivity $c(\lambda)$ of the respective bands of the TCS3472 sensor (Figure 2), to achieve intercomparable results:

$$T_{R,G,B,C} = \frac{\int E_{trans}(\lambda) \cdot c_{R,G,B,C}(\lambda) \, d\lambda}{\int E_{in}(\lambda) \cdot c_{R,G,B,C}(\lambda) \, d\lambda}$$

A time series of light transmittance for the different spectral channels and both instruments is shown in Figure 10a. Ice

transmittance in the clear channel decreased from 3% at the start of measurements to $< 2$ % before the failure of the bottommost sensors. Transmittance derived from the light chain is generating a time series consistent to the RAMSES measurements. Sea ice transmittance is slightly overestimated in the clear and the blue channel, while the light chain underestimates transmittance in the red and green channel. Overall, the agreement between both setups is striking, with root mean square errors (RMSE) of 0.003, 0.0048, 0.0036, 0.0059, respectively for the red, green, blue and clear channels. A

maximum RMSE of 0.59% transmittance is a particularly good result, given the above mentioned geometric limitations, and a cost reduction by more than a factor of 10.

### 3.6    Spectral signatures

The four spectral bands of the light sensor chain also allow a simple assessment of light color and spectral changes over time. The RGB rendering of light chain data in Figure 11 clearly shows a change in light color from white sunlight at the

surface to a blueish color in the ice interior. Even some reddish hues of skylight before arrival of the polar night are apparent in the data. These first results suggest that there is great potential to detect at least high concentrations of in-ice algae by this light sensor chain, either in RGB plots or simple band ratios similar to remote sensing algorithms. Unfortunately, a total





failure of our buoy in December 2018, far before the first major spring blooms, prevents us from presenting such an analysis here.

To assess the radiometric quality of multispectral light chain data, we also compared RAMSES measured light spectra with the corresponding chain derived transmittance data for individual days (Figure 12). A consistent under-estimation of chain measurements in the green makes it however difficult to derive a true spectral shape of transmitted from light chain measurements. This might be caused either by an overall low spectral accuracy of the low-cost sensors, or a spectral sensitivity that differs from the one provided by the sensor manufacturer. These issues could certainly be addressed by a

more detailed individual spectral calibration of all chain sensors, which in turn would however jeopardize our low-cost approach. High spectral accuracy can be achieved on classical radiation station setups using for example the RAMSES hyperspectral radiometers [*Nicolaus et al.*, 2010b] and thus does not need to be provided by the light sensor chain. The low-cost approach of the light-sensor chain however allows for much more widespread deployments, with the potential to yield a much better spatio-temporal resolution of sea-ice associated light measurements in the Arctic and Antarctic.

Apart from shadowing and uncertainties in the calibration and spectral response, the differences in the above comparisons between the light sensor chain and the RAMSES measurements might also arise from the different measured basic irradiance quantities, namely planar and sideward planar/ scalar irradiance, and thus slightly different spectral signatures. For detection of spectrally distinct features, such as ice-algal blooms, spectral attenuation coefficients as well as band ratios are thus more useful than absolute spectral fluxes.

**4    Discussion**

**4.1    Implications for radiative transfer modeling**

Most autonomous optical measurements in the sea ice environment have been limited to measurements above and below the ice [*Nicolaus et al.*, 2010a; *Wang et al.*, 2014]. Detailed investigations show however, that a vertically resolved measurement provides a better basis for the estimation of light attenuation in sea ice [*Ehn et al.*, 2008a; *Light et al.*, 2008]. In our case, the

light sensor chain allows for a direct autonomous measurement of the diffuse attenuation coefficient within the ice interior. Equivalency of scalar and planar diffuse attenuation coefficients within the asymptotic regime in the ice interior make this direct measurement possible. The acquisition of such data is urgently needed as input parameter for simple radiative transfer schemes in large scale models.

Our results also highlight the strongly non-exponential decay of scalar irradiance in the lowermost 50 cm of the ice cover.

This effect is currently unaccounted for in simple exponential radiative transfer parameterizations, but is significantly reducing light levels at the ice bottom. Exponential models of the decay of downwelling planar irradiance have a much wider vertical range of applicability, but also here photon loss in the lower ice portion is currently unaccounted for. A precise measurement of the in-ice diffuse attenuation coefficient however allows at least for accurate tuning of exponential parameterizations. This is of great advantage, as this parameter cannot be easily derived from the inherent optical properties





of the ice. To test the accuracy of such a measurement supported exponential parameterization, we compare the sea ice transmittance time-series acquired by the RAMSES hyperspectral setup to a simple exponential parameterization, where sea ice light transmittance $T$ is parameterized as a function of ice thickness $z$, ice surface albedo $\alpha$ as measured by the RAMSES and chain derived ice-interior diffuse attenuation coefficient $\kappa_{chain}$:

$$T = (1 - \alpha) \exp(-\kappa_{chain} \cdot z)$$

The resulting time series (Figure 13) is in close agreement with the light transmittance as measured by the RAMSES setup.
Measured albedo values were reduced by 0.05 to avoid albedo values larger than 1.

### 4.2 Limitations

As mentioned above (section 2 and 3.1), the radiometric quality of the sensors after initial field calibration is sufficient for studies of the light field in all four spectral channels. While the clear channel is very close in spectral characteristics to commercially available sensors for photosynthetically active radiation (PAR) it is cutting off already at 650nm instead of
700nm (Figure 2). This does not have a huge impact on flux measurements under sea ice, but slightly overestimates PAR transmittance.

The absolute spectral accuracy of the presented system is unfortunately limited, as shown in section 3.6. However it is still highly useful for the observation of temporally changing band ratios. Similarly, this low-cost system does not provide a true measurement of sea ice transmittance of planar irradiance, as the sideward-looking sensors can introduce artefacts due to
azimuthal orientation and self-shadowing.

The geometry of the sideward-looking setup poses some challenge for the interpretation of data from the light sensor chain. This was however solved by the equivalency of sideward planar irradiance and total scalar irradiance, as total scalar irradiance is a primary target for biological studies. In addition, total scalar irradiance can be easily converted back to planar irradiance in the case of a known mean cosine of the light field. Our model results together with recent other works [Matthes
et al. 2019 [*Katlein et al.*, 2014b] can provide guidance on the appropriate choice of mean cosine in and underneath sea ice.

The detection limit of the sensor is sufficiently low to provide reliable optical data at absolute irradiances down to less than 0.05 W/m². This enables the light chain to detect light levels under two meter thick Arctic ice in autumn, e.g. even when the sun is already a few degrees below the horizon. This dynamic range can be further increased with more advanced settings on sensor gain and exposure.
The impact of the deployment hole itself and its refreezing process is however of even less importance than for traditional thermistor string observations. The sideward planar irradiance sensors observe scattered photons from a larger footprint and are thus not strongly influenced e.g. by the distance of the sensor to the hole wall. In turn, larger brine channels, or air inclusions can locally alter the light field, leading to the observed scatter in the retrieved vertical attenuation coefficients.

While the sensor spacing of 5 cm seems to excellently resolve the vertical decay of light within sea ice, this vertical
resolution is not high enough to decipher detailed optical properties of the snow pack. Also the precise detection of the





vertical position of interfaces between water, ice, snow and air is limited. These measurements can however be easily obtained from co-deployed e.g. sonic ice and snow thickness sounders.

### 4.3 Recommendations for future deployments

For future deployments of the system, we want to stress the importance of sensor calibration before deployment. This can be simply done by stretching the chain out e.g. along the ships railing and fixing it in a place with comparable lighting conditions all along the chain. To achieve best calibration accuracy, this exercise should be performed for several days and in the best case during foggy weather with a close to isotropic light field. If this step is omitted, radiometric accuracy of the light sensor chain is much reduced.

Also, it should be noted that sufficient information about the ice interior attenuation coefficient – and thus the ice inherent optical properties – can only be achieved if the light sensor chain is deployed on sufficiently thick ice, where the asymptotic limit is reached. Our data suggests that this condition will not be satisfied for summer sea ice with a thickness below one meter. While the sensors will still provide reliable radiometric measurements, a detailed interpretation of attenuation coefficients then needs to be supported by explicit radiative transfer modelling.

A great opportunity will be the co-deployment of light sensor chains with classical thermistor-chain buoys. This will allow for a closer investigation of the dependency of ice optical properties and the thermodynamic state of the ice. It might also provide crucial input for structural optical models and radiative transfer parameterizations. Especially the low unit cost enables more frequent autonomous deployment of optical sensors to cover some aspects of spatial variability. A combination of light-chain deployment with ice core analysis of temperature, salinity and texture close to the deployment site will further improve the interpretation of the optical data.

For the next deployments, an upgraded version of the sensor chain will provide more physical strength, to avoid partial chain failures as observed in this first deployment. Furthermore, the light sensor chain will be outfitted with a standard serial interface enabling easy integration in almost any data logging system.

To increase the scientific use of the sensor chain especially during the dark period of the year, the chain can be upgraded to contain LED lighting elements. This would allow for active sensing of the ice optical properties also in the winter. If the spatial resolution of optical sensors along the chain is increased, this chain could also be used for diffuse reflectance spectroscopy [*Kim and Wilson*, 2011], where the inherent optical properties of sea ice can be determined by active measurement of the point spread function in sea ice [*Maffione et al.*, 1998; *Voss et al.*, 1992].

### 5 Summary

We deployed and evaluated a first prototype of an in-ice light sensor chain, designed on the basis of previously developed digital thermistor chains. With only minor sensor failures, the device acquired optical data throughout the autumn of 2018 in the vicinity of the geographic North Pole. Overall we could show that this newly developed sensor chain is able to provide valuable autonomous measurements of the light field within sea ice. The four-channel multispectral sensors embedded in the chain allow for the measurement of a vertical profile of PAR total scalar irradiance, as well as retrieval of multispectral





diffuse attenuation coefficients particularly in the ice interior. The equivalency of total scalar and sideward planar irradiance
is significantly helping the interpretation of the acquired measurements. Measurements within the asymptotic regime in the
ice interior allow a more direct relation of optical measurements to the inherent optical properties of sea ice than traditional
measurements above and below the sea ice. In addition, the low cost-factor of the presented system will allow for more
frequent deployments, which will enable us to achieve a much better spatial and temporal coverage of ice-associated light
data and help to better understand in particular the spatial variability in sea-ice optical properties.

## 6 Acknowledgments

Development and deployment of the unit, as well as the positions of CK and MH were funded by the Helmholtz
Infrastructure Initiative "Frontiers in Arctic marine Monitoring (FRAM)" and the Alfred-Wegener-Institut Helmholtz-
Zentrum für Polar- und Meeresforschung. Scientific data evaluation and writing of the manuscript was supported by a
Sentinel North Postdoctoral Research Fellowship to CK and the Takuvik Joint International Laboratory (Université Laval
and CNRS). This work represents a contribution to the Diatom ARCTIC project (NE/R012849/1; 03F0810A), part of the
Changing Arctic Ocean programme, jointly funded by the UKRI Natural Environment Research Council (NERC) and the
German Federal Ministry of Education and Research (BMBF).

Deployment of the system and participation in the AO18 expedition of the Swedish icebreaker Oden was facilitated by the
Swedish Polar Research Secretariat (SPRS). We want to thank in particular Philipp Anhaus and Matthieu Labaste for field
assistance, as well as Anja Nicolaus and Marcel Nicolaus for administrative support around the buoy deployment.

## 7 Author Contributions

CK had the idea of a digital light sensor chain, wrote the first draft of this manuscript, processed and evaluated the light
chain data. LV and his team built the prototype unit in close collaboration with MH and CK. MH and CK deployed the
instrument in the field. SLG and CK performed the radiative transfer modelling of the sideward irradiance. All authors
contributed to the editing of the manuscript.

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



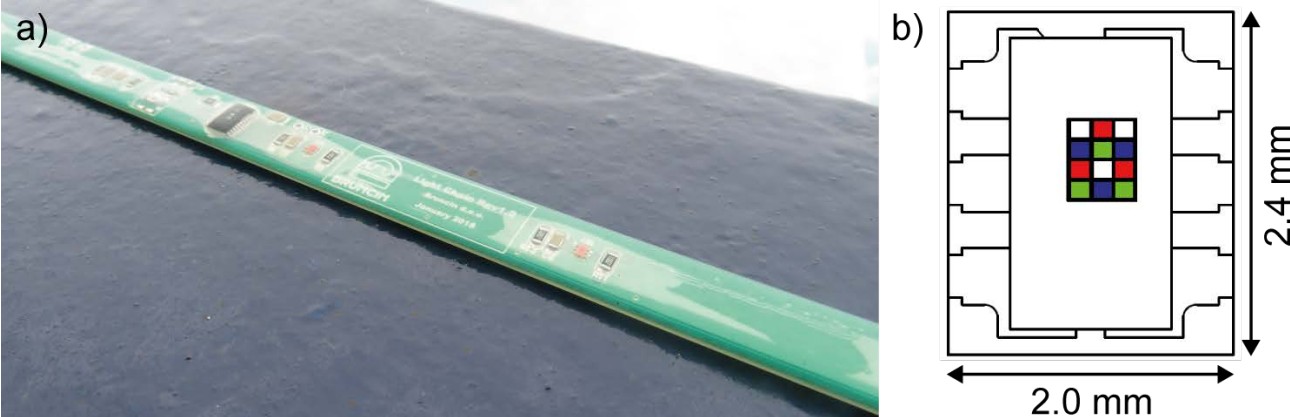


**Figure 1: a) a segment of the chain prototype during the calibration procedure. The optical sensors are visible as red-colored components on the circuit board. The black addressing chip controlling the sensors of a particular chain section can be seen in the middle of the picture. b) Sketch of the arrangement of the different spectral channels on the 3x4 photodiode array of the sensor.**

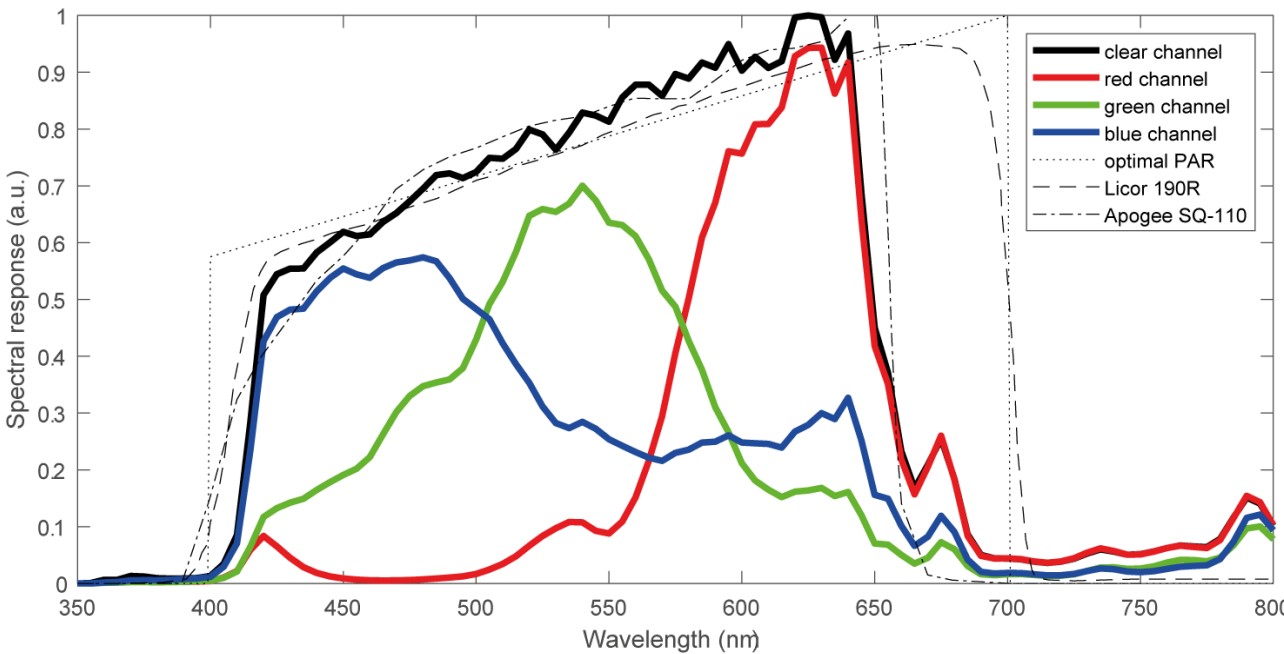

**Figure 2: Spectral sensitivities of the red, green, blue and clear channel (solid lines). Thin black lines show the spectral sensitivities for different kinds of PAR sensors: theoretical PAR response (dotted line), Licor 190R sensor (dashed line) and Apogee SQ-110 silicon photodiode sensor (dash dotted line).**





**Figure 3: a) Picture of the radiation station setup few days after deployment on 23 August 2018: From left to right one can see the two cases with electronics, separate batteries and a webcam, the tripod holding a snow height sensor and the thermistor chain, the radiation frame holding the Ramses radiometers and the small tripod holding the light chain. b) close up of the light chain. c&d) The system state on 15 September 2018 e) Under-ice RAMSES sensor measuring transmitted hyperspectral irradiance. f) Lowermost portion of the light sensor chain protruding the ice-ocean interface. Note the attached metal weight to keep the chain vertical in the water column.**





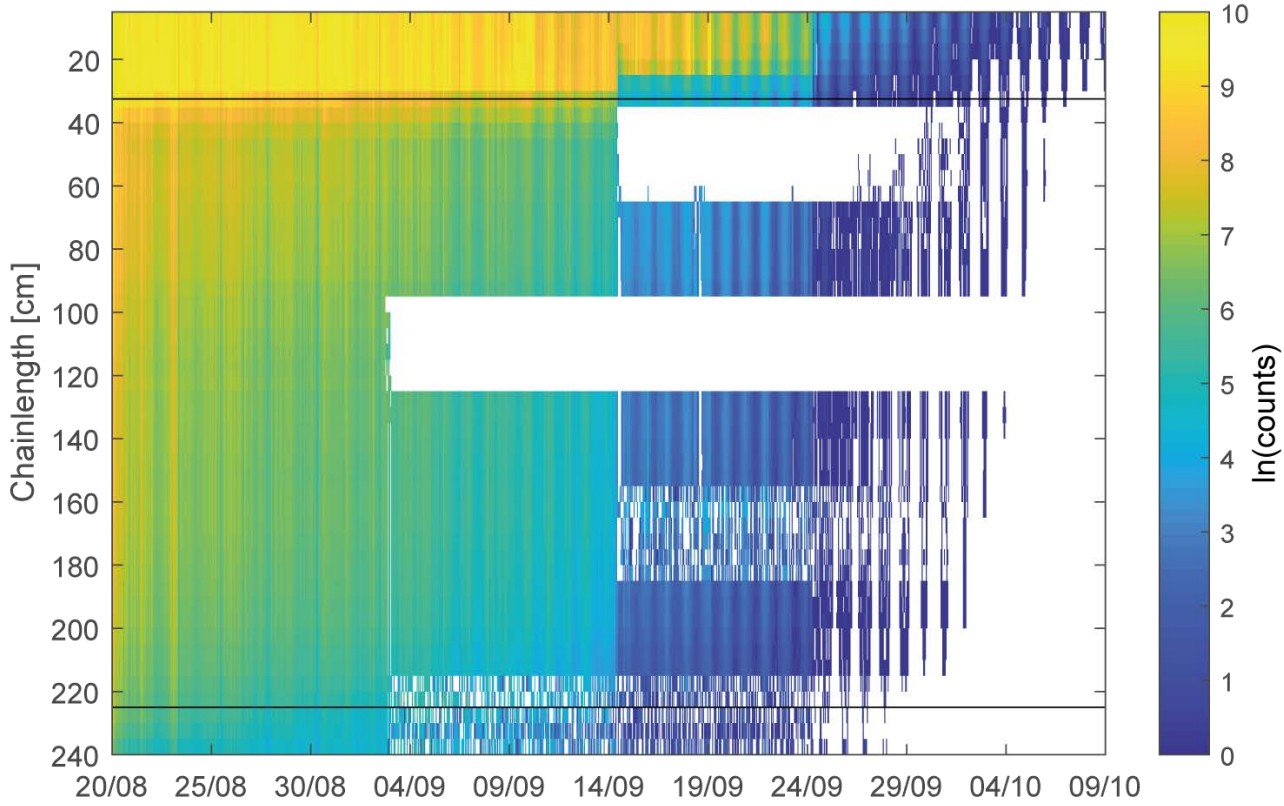

**Figure 4: Data from the clear channel of the light chain, shown as logarithm of sensor count after cross-calibration of chain sensors. Horizontal black lines indicate the air-ice, as well as the ice-ocean interface during deployment of the light chain.**






**Figure 5: Vertical profiles of sideward planar irradiance through sea ice: a) Irradiance decay in the four spectral channels shortly after deployment on 20 August 2018. b) Temporally increasing light attenuation for the clear channel from August 20 to September 2. Black lines indicate the interfaces between air, ice and ocean.**





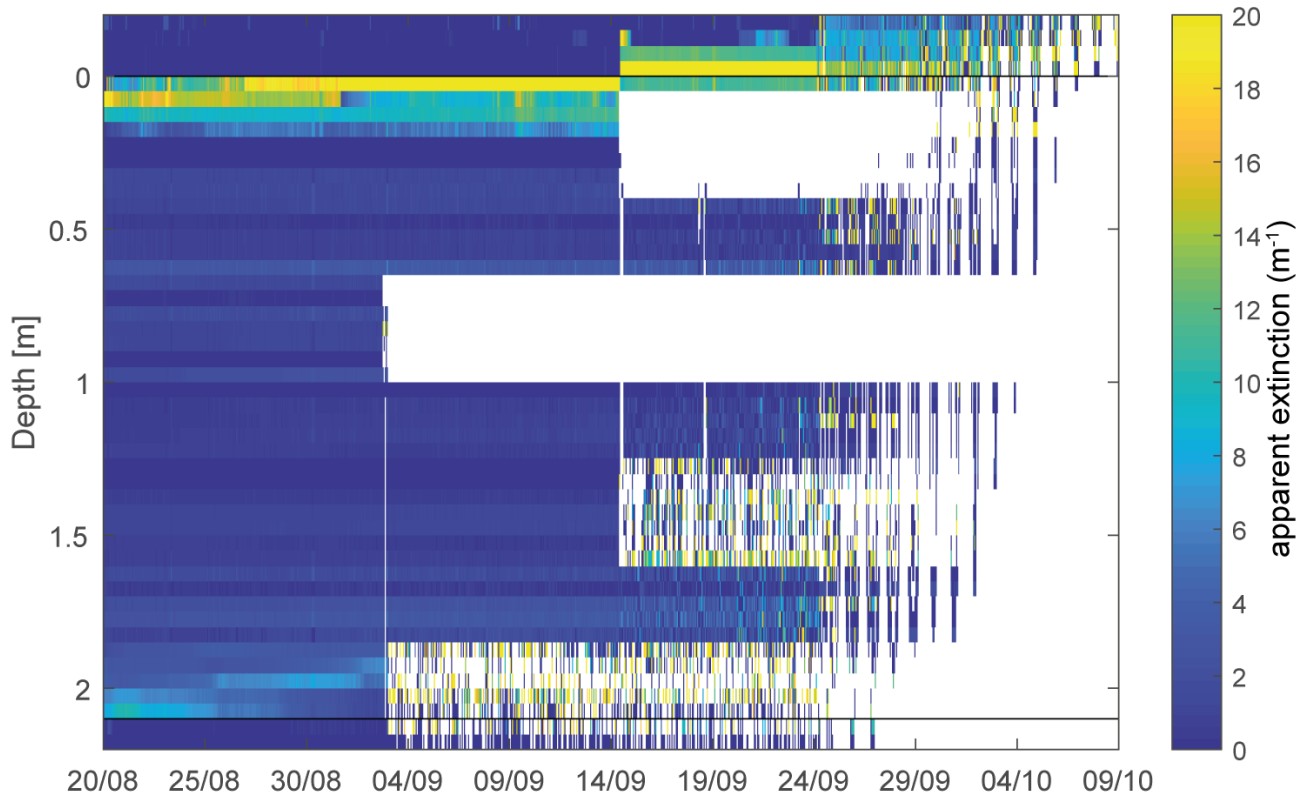

Figure 6: Vertical profiles of attenuation coefficients calculated from light chain data. Black lines indicate the interfaces between air, ice and ocean during chain deployment.





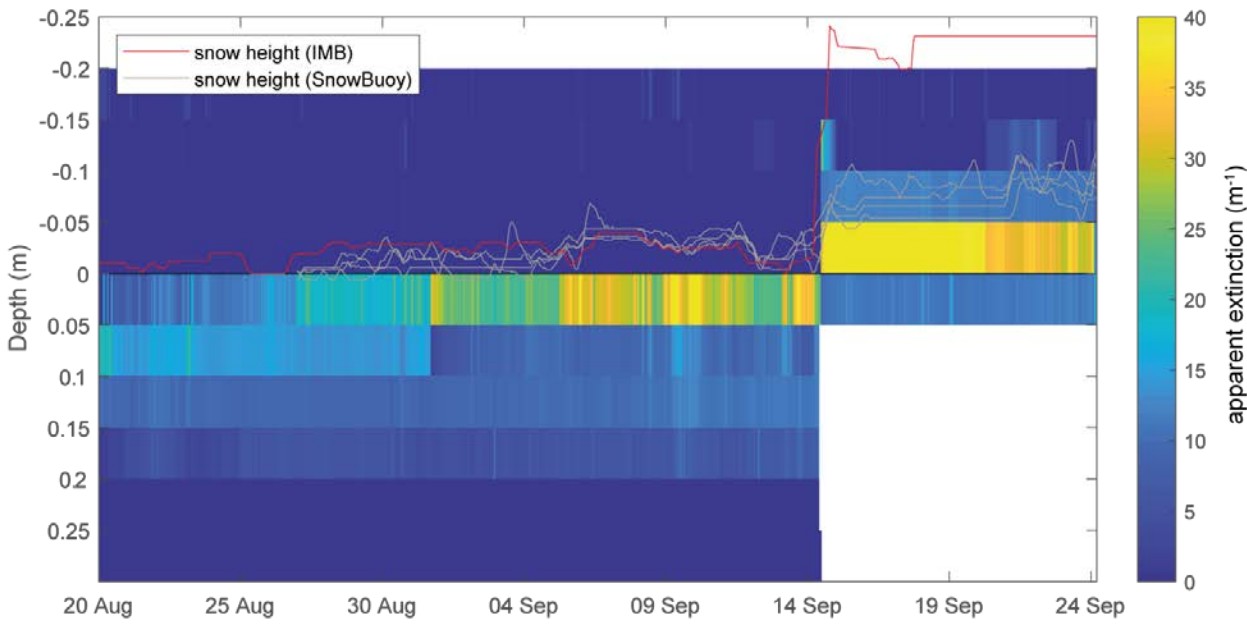


**Figure 7: Close up of the vertical profiles of attenuation coefficients calculated from light chain data in the uppermost sensor pairs. Snow height measured from the neighboring acoustic pinger of the ice-mass-balance buoy (red) and from the four acoustic pingers of the snow buoy (gray lines). The black line indicates the interfaces between air and ice during chain deployment.**





Figure 8: a) Time-averaged attenuation coefficients for the first complete set of time series until 14 September for all four channels. b) Bulk attenuation coefficient averaged over the entire vertical domain. From top to bottom: red, green, blue, clear channel. c) Ice interior attenuation coefficient for all four channels, determined as average of attenuation coefficients from 0.35 to 1.8 m depth. d) Time series of clear channel attenuation coefficient for different vertical ice layers. The black line equals the black line in c. Sudden changes on 3 September 2020 are caused by partial failure of the sensor chain.






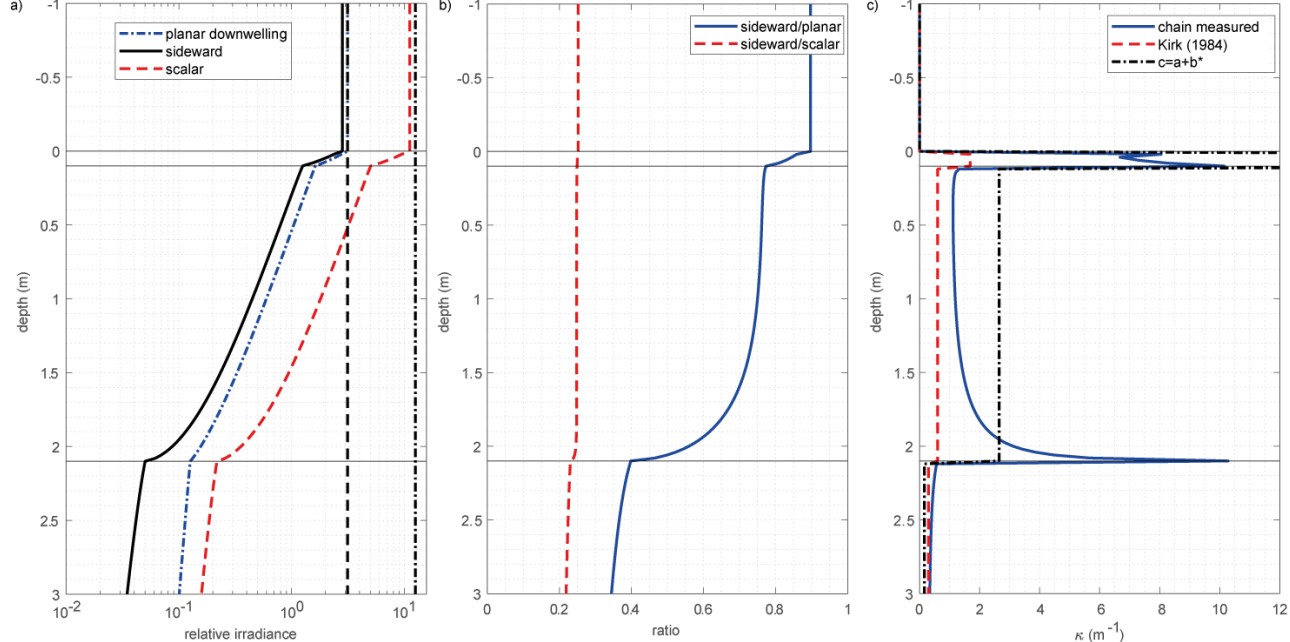

**Figure 9: a) Modeled irradiance curves for planar downwelling irradiance (blue dash dotted line), sideward irradiance (black solid line) and total scalar irradiance (red dashed line). Vertical black lines depict the values of $\pi$ (dashed line) and $4\pi$ (dash dotted line), representing the values of planar downwelling and total scalar irradiance, respectively. Horizontal black lines indicate the interfaces between atmosphere, surface scattering layer, ice interior and water. b) Modeled ratios between sideward and planar**
**downwelling (blue solid line), as well as sideward and total scalar irradiance (red dashed line). c) Modeled attenuation coefficients as measured by the chain (blue solid line), as well as pseudo-IOP attenuation coefficients derived from model IOP: diffuse attenuation coefficient parameterized according to *Kirk* [1984] (dashed red line) and the beam attenuation coefficient (dash dotted black line).**






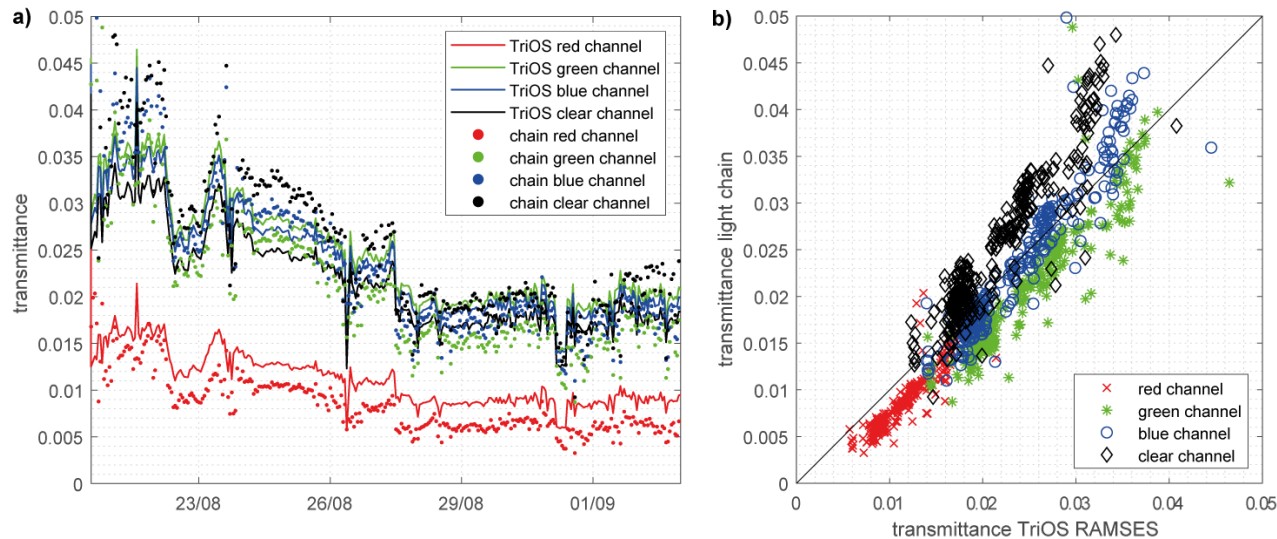

**Figure 10: a) Time series of sea ice transmittance in four spectral bands derived from top and bottommost chain sensors (dots), as well as from TriOS Ramses radiometers above and below the ice (solid line). b) Scatter plot of the dataset showing how close measurements from the light chain agree with high quality hyperspectral spectroradiometers.**

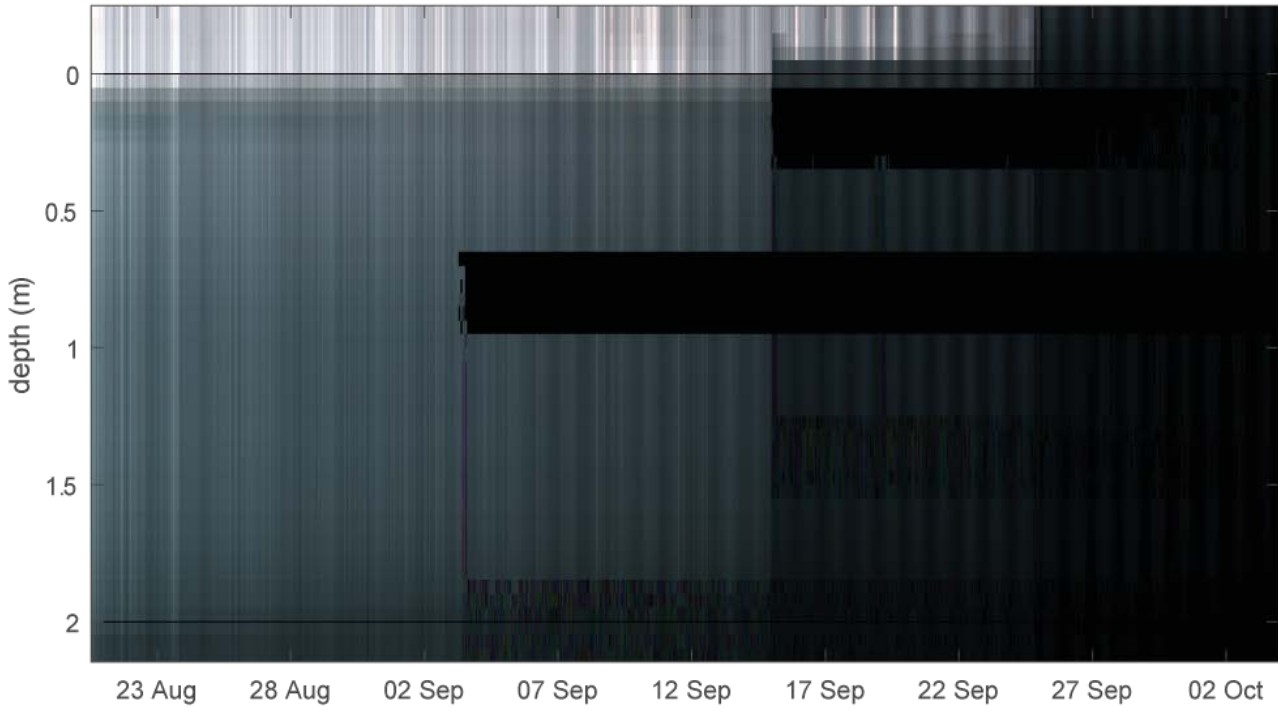


**Figure 11: RGB rendering of chain data using a gamma function with $\gamma = 0.3$.**

**Figure 12: Spectral transmittance data of the five measurements centered around solar noon during nine days of the deployment: Transmittance spectra measured by the TriOS RAMSES radiometers (yellow) are overlain by horizontal lines for the four different spectral bands (red, green, blue, clear) reconstructed from TriOS RAMSES spectra (solid lines) and chain measurements (dashed lines).**




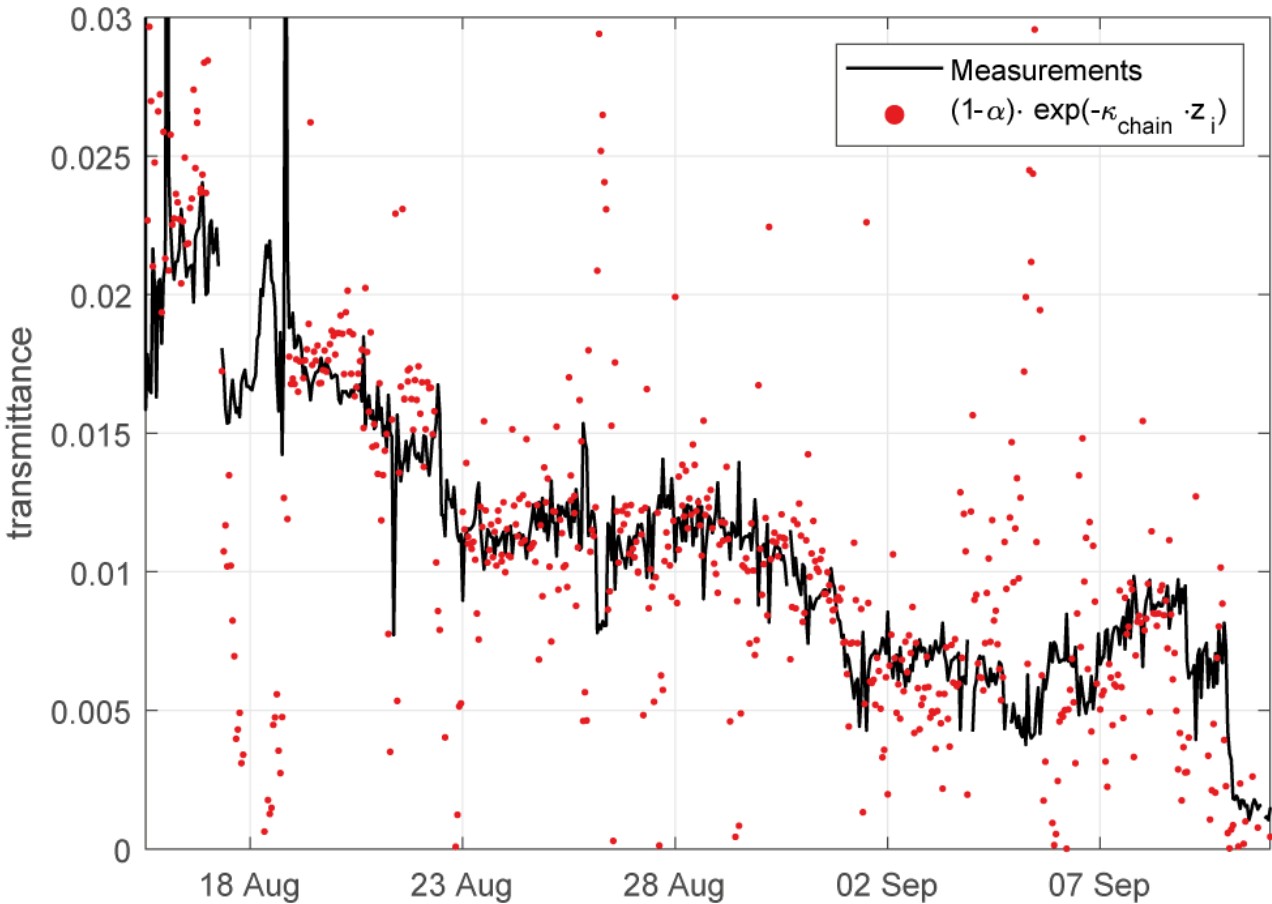

**Figure 13: Time series of sea ice light transmittance as observed by the traditional RAMSES setup (black line) and a simple light chain data assisted exponential parameterization (red points) based on observed albedo and diffuse attenuation coefficients. The high scatter of the parameterizations is caused by scatter in the albedo data measured by the RAMSES setup.**