# Peer review of "New insights into radiative transfer within sea ice derived from autonomous optical propagation measurements"

_The Cryosphere, 2020_

## Referee Comment (RC1) · Anonymous Referee #1 · 3 Sep 2020

This manuscript describes the development and implementation of a novel sensor system for the measurement of shortwave radiation within ice. The "light chain" is a simple, inexpensive, easy-to-deploy instrument that collects optical propagation data within a 5 cm diameter bore hole autonomously. The demonstration deployment was in ∼2m thick sea ice in the vicinity of the North Pole. This tool is novel and, I expect, will be very useful for understanding the propagation of light through ice, a topic which is highly relevant to current climate research. Beyond development and implementation, the manuscript offers insights regarding the transport of light within the ice in ways that could fundamentally streamline many routine measurements. I find this paper to be very nicely written, easy to read, appropriately referenced, and supported by clear

illustrations and informative figures.

Overall, the manuscript gives a great overview of the motivation, methods, and materials of this new system. I am pleased to see the conclusion regarding the proportionality that exists between the side-welling planar irradiance and the spherical irradiance. This seems to be a very useful result. I have only a few minor questions and a few technical points:

The title is fine, and it is completely acceptable to leave as is. However, I suggest a modification: 'New insights into radiative transfer within sea ice derived from autonomous optical propagation measurements' might be slightly more informative?

It appears the data from this system were perhaps downloaded locally (during the 4 weeks the ship was on station)?, but then telemetered (past September), but this is never explicitly stated. It would be helpful to know what the telemetry requirements look like.

Line 15: shortwave shouldn't be hyphenated

Line 115: "The ice was homogeneously grown,..."? Please clarify what is meant by this statement.

Line 155: "in the clear"? does this refer to the broadband channel on the sensor? It's not obvious.

157: delete "both,"

Fig 6 caption: "attenuation coefficients", color bar label: "apparent extinction". It would be helpful if the terminology was consistent.

Fig 7: I am assuming that the borehole did not immediately re-freeze, given the August deployment, but it would be helpful to know what that process looked like? I wonder if some of the features shown in Fig. 7 are associated with the refreezing process? In particular, I would not expect the surface scattering layer to re-form within the bore

[Figure]

hole after drilling and installation. Why does a highly scattering surface layer appear to increase so quickly between 31 Aug and 14 Sept—maybe that is a SSL reestablishing? It is interesting that the attenuation in the uppermost 5 cm of the ice drops so dramatically when snow began to accumulate. I suppose that happens because the uppermost portion of the ice is no longer at the top boundary, and the new snow above is now attenuating light strongly. Should one be surprised that this attenuation drops so much?

Fig 8b shows values increasing 1.7 to 2.1m-1. Is this really an ice-evolution time series? Or a refreezing bore hole time series?

247: not "identical", but "proportional"!

Fig11: I need a bit more info to know how to look at this RGB rendering. I don't see obvious colors, but perhaps some guidance could help?

Fig 12 (and line 286): looks like the Trios transmittance data are picking up some chla absorption (strong dip wavelengths < 470 nm)? Is this detectable in the light chain data?

315-316: significantly reducing? Please quantify!

361: data suggest (not suggests); also...why only summer?

---

## Referee Comment (RC2) · Anonymous Referee #2 · 2 Oct 2020

This manuscript presents an interesting and potentially very useful new design for measuring the light field within sea ice. Making use of relatively inexpensive light sensors, the authors sacrifice some degree of accuracy in the measurements for the ability to have many, closely spaced instruments on the light string, which can be left behind after deployment. The low cost and ease of deployment will also allow for installation of the strings at many locations, similarly to the thermistor strings used as inexpensive mass balance buoys.

The concept is well presented and the first results are analyzed and interpreted in a way that shows the instrument's strengths and weaknesses. I feel it warrants publication in

[Figure]

The Cryosphere. The user community would likely benefit from a supplement that gives more technical details of the design.

One of the results the authors present that is important for the ability to use the side-looking irradiance measurements in more traditional applications is that, at least with diffuse incidence, the sideward looking irradiance is proportional to the scalar irradiance, a very useful quantity for biology in the ice and ocean. This result is relied upon as a justification for the unusual orientation of the irradiance sensors on the string. While it may well be an accurate and useful result, I felt the presentation and examination of the modelling this result is based on is the weakest part of the paper and should be improved before final publication. If this tool becomes broadly used, this result will be widely used and cited, so it should be well founded here. Beyond that my comments below are mostly minor.

Regarding the modelling and interpretation of the model results:

1. It appears there was no attempt to run the model for different wavelengths, given that the absorption coefficient is fixed. I feel it would be worth confirming that the factor-of-four difference to spherical irradiance holds at other wavelengths

2. How were the absorption and scattering coefficients and asymmetry parameter (a/b/g in §2.5) chosen? No references or reasons are given for the chosen values. Based on the estimates from Light et al (2008, 10.1029/2006JC003977), b=250 m-1 seems low for the surface scattering layer (SSL), though b=25 m-1 is in their range for interior ice. Warren et al (2006, 10.1364/AO.45.005320) give much lower values of a for pure ice. Is a=0.15 based on measurements of sea ice? Finally, g=0.9 is very low according to Light et al, who argued that the value they used, 0.94, "is probably too low for the majority of actual g values appropriate for sea ice."

3. The model has isotropic downwelling radiance above the ice, which is a reasonable approximation under thick clouds, which admittedly describes the high Arctic most of the summer. Still, it would be useful to see if and at what depths this relationship holds

with a direct beam, something which will be azimuthally dependent and potentially spectrally dependent.

4. Scattering obviously dominates in the SSL, so it might not matter, but was the density used to account for the fact that the air in the SSL reduces the absorption since only 40-50% of the path is ice?

5. Showing some comparisons of the modelled normalized irradiance and extinction coefficient with the observed values would give readers more confidence in relying on the model result giving the proportionality between sideward and scalar irradiance.

Other comments:

6. Line 77: Was the transparent heat shrink checked to make sure it was spectrally neutral, or could it have altered the response curves shown in Figure 2?

7. Line 79: 48 sensors spaced 5 cm apart gives a total chain length (from first sensor to the last) of 235 cm.

8. Line 95: Presumably one of the 'eight's should be 'six'.

9. Line 155: I would write 'channel' after 'clear' since 'in the clear' is an expression.

10. While the boundary effects at the bottom of the ice are explained in section 3.1, the increase in irradiance between 0.2 and 0.3 m, just below the SSL, is not mentioned. It should at least be pointed out and possible reasons for it discussed. Is it an effect of the hole, or are the lower sensors seeing more of a nearby bare area or pond? It could also be a real effect of the boundary. The light will refract on entering the ice from the air in the SSL, meaning it will wind up being more downward directed just below the surface, with very little light travelling nearly horizontally (since that would be reflected at the ice surface), so perhaps a side view a few cm below the air(SSL)-ice boundary is actually darker than after some scattering in the ice. I'm not sure you can come with a definitive answer for why the measured radiance increases in that layer, but it shouldn't be ignored.

11. Line 195: Figure 5 doesn't show any data after the 14 September snowfall, so it is illustrating exponential decay within the SSL.

12. The paragraph starting at line 215 should address why the clear channel has lower bulk extinction than any of the others, rather than being between the R and the G/B channels.

13. I would re-write line 225-226 to read 'with attenuation coefficients for the clear channel rising from 0.8 m-1 to 1.0 m-1 and for the red from 1.1 m-1 to 1.3 m-1.'

14. Figure 3 is not referenced in the text.

15. Figure 4: I would specify in the caption that it is the natural logarithm of sensor count that is plotted.

16. I think Figure 6 would be more readable with a logarithmic scale on the color axis, or separate color bars put together to allow seeing the variation within the ice. The inclusion of Figure 7 allows for seeing details at the surface.

17. Since the focus of Figure 11 is the colour of the light, it might work better to use a constant brightness. When I look at it, on screen or paper, the colour information is largely drowned out by the brightness variations.

---

## Author Comment (AC1) · 3 Nov 2020

**Replies to reviewer comments**

Please find our answers to the reviewer's overall minor comments in blue font under the respective comments.
We thank you very much for the constructive reviews and the interest in publishing our work in The Cryosphere.

**Anonymous Referee #1**

This manuscript describes the development and implementation of a novel sensor system for the measurement of shortwave radiation within ice. The "light chain" is a simple, inexpensive, easy-to-deploy instrument that collects optical propagation data within a 5 cm diameter bore hole autonomously. The demonstration deployment was in~2m thick sea ice in the vicinity of the North Pole. This tool is novel and, I expect, will be very useful for understanding the propagation of light through ice, a topic which is highly relevant to current climate research. Beyond development and implementation, the manuscript offers insights regarding the transport of light within the ice in ways that could fundamentally streamline many routine measurements. I find this paper to be very nicely written, easy to read, appropriately referenced, and supported by clear illustrations and informative figures.
We thank you very much for the positive evaluation of our manuscript and are happy that we could convince you of the usefulness of our approach.

Overall, the manuscript gives a great overview of the motivation, methods, and materials of this new system. I am pleased to see the conclusion regarding the proportionality that exists between the side-welling planar irradiance and the spherical irradiance. This seems to be a very useful result. I have only a few minor questions and a few technical points:
We thank you very much for your careful evaluation and appreciate your constructive comments.

The title is fine, and it is completely acceptable to leave as is.   However, I suggest a modification: 'New insights into radiative transfer within sea ice derived from autonomous optical propagation measurements' might be slightly more informative?
We revised the title accordingly.

It appears the data from this system were perhaps downloaded locally (during the 4weeks the ship was on station)?, but then telemetered (past September), but this is never explicitly stated.  It would be helpful to know what the telemetry requirements look like.
We added the following sentence to section 2.1: "Measured data were sent via an Iridium SBD satellite link requiring data transfer of around 65kB per day for the hourly sampling schedule."

Line 15: shortwave shouldn't be hyphenated
Corrected accordingly

Line 115:  "The ice was homogeneously grown,..."?  Please clarify what is meant by this statement.

We removed this ambiguity and it now reads "The level ice was covered by a 10-15cm thick surface scattering layer."

Line 155: "in the clear"? does this refer to the broadband channel on the sensor? It's not obvious.
We added the word "channel" for clarity.

157: delete "both,"
deleted

Fig 6 caption: "attenuation coefficients", color bar label: "apparent extinction". It would be helpful if the terminology was consistent.
We changed the Figure caption to "apparent extinction coefficients" also for figure 7

Fig 7: I am assuming that the borehole did not immediately re-freeze, given the August deployment, but it would be helpful to know what that process looked like? I wonder if some of the features shown in Fig. 7 are associated with the refreezing process? In particular, I would not expect the surface scattering layer to re-form within the bore hole after drilling and installation. Why does a highly scattering surface layer appear to increase so quickly between 31 Aug and 14 Sept maybe that is a SSL reestablishing? It is interesting that the attenuation in the uppermost 5 cm of the ice drops so dramatically when snow began to accumulate. I suppose that happens because the uppermost portion of the ice is no longer at the top boundary, and the new snow above is now attenuating light strongly. Should one be surprised that this attenuation drops so much?
To arrive at realistic observations of the topmost cm, the top of the hole was backfilled with cuttings and the surface scattering layer restored. While this is not exactly the natural configuration, it is extremely close to it. This backfilling also avoids any light leaking through the top of the hole, such that influence of the hole during refreezing is minimal due to the large sensor footprint.
To specify this, we added "The topmost part of the hole was backfilled with cuttings and the surface scattering layer restored." to the description of the deployment in section 2.3.
Thus features visible in Figure 7 are not related to the refreezing of the hole, but only the deposition of snow, where a few mm have dramatic effects on the optics increasing light extinction e.g. between 31 Aug and Sep 14. (See section 3.3). We added a statement "Spectrally integrated attenuation in the layer directly beneath the surface is largest even when the interface location changes." To refer to the "drop" in attenuation which we are not surprised about (e.g. Grenfell & Maykut 1977). In general it should be noted that data at the surface can't be overinterpreted, as the spatial resolution with 5cm sensor spacing is too crude as we mention already in section 4.2 "While the sensor spacing of 5 cm seems to excellently resolve the vertical decay of light within sea ice, this vertical resolution is not high enough to decipher detailed optical properties of the snow pack and surface scattering layer. Also the precise detection of the vertical position of interfaces between water, ice, snow and air is limited."

Fig 8b shows values increasing 1.7 to 2.1m-1. Is this really an ice-evolution time series? Or a refreezing bore hole time series?

As we see these increases in all depth layers, we do not suspect this to be a result of hole refreezing and are confident that this is really an ice evolution time series as stated.

247: not "identical", but "proportional"!
Corrected accordingly

Fig11: I need a bit more info to know how to look at this RGB rendering. I don't see obvious colors, but perhaps some guidance could help?
As both reviewers did not seem to understand this figure and we did not manage to generate a clearer representation, we deleted this figure and rephrase the respective section accordingly: "The four spectral bands of the light sensor chain also allow a simple assessment of light color and spectral changes over time. Our first results suggest that there is potential to detect at least transient high concentrations of in-ice algae by this light sensor chain, either in RGB plots or simple band ratios similar to remote sensing algorithms."

Fig 12 (and line 286): looks like the Trios transmittance data are picking up some chla absorption (strong dip wavelengths < 470 nm)? Is this detectable in the light chaindata?
As the strength of this dip doesn't seem to change much over time, there is also no clearly evident chlorophyll signal in this light chain dataset. From recent deployments this year (not to be described here) we however know that we can detect chlorophyll signals. Due to the limited spectral accuracy it is easier to detect transient chlorophyll signals by the light chain. We added this to the respective section.

315-316: significantly reducing? Please quantify!
We deleted the word significantly and provided a rough quantificantion of this effect from the measurements: "This effect is currently mostly unaccounted for in simple exponential radiative transfer parameterizations, but is reducing light levels at the ice bottom, by up to a factor of 2-3" Some exponential models might have been tuned to use 'inaccurate' extinction coefficients yielding correct results in turn.

361: data suggest (not suggests); also...why only summer?
Corrected accordingly and removed the word summer.

---

## Author Comment (AC2) · 3 Nov 2020

**Replies to reviewer comments**

Please find our answers to the reviewer's overall minor comments in blue font under the respective comments.
We thank you very much for the constructive reviews and the interest in publishing our work in The Cryosphere.

**Anonymous Referee #2**

This manuscript presents an interesting and potentially very useful new design for measuring the light field within sea ice. Making use of relatively inexpensive light sensors, the authors sacrifice some degree of accuracy in the measurements for the ability to have many, closely spaced instruments on the light string, which can be left behind after deployment. The low cost and ease of deployment will also allow for installation of the strings at many locations, similarly to the thermistor strings used as inexpensive mass balance buoys.
We thank you very much for the positive evaluation of our novel measurement concept.

The concept is well presented and the first results are analyzed and interpreted in a way that shows the instrument's strengths and weaknesses. I feel it warrants publication in The Cryosphere. The user community would likely benefit from a supplement that gives more technical details of the design.
Most technical details of the design are given in in sections 2.1-2.3. Thus it is unclear to us what you are referring to. This publication does not aim to publish completely documented open source hardware including PCB drawings and parts lists. However we are confident that any interested user with access to the respective facilities can easily built is own version of this chain. Also the manufacturer has a fairly open policy regarding making their designs available.

One of the results the authors present that is important for the ability to use the side-looking irradiance measurements in more traditional applications is that, at least with diffuse incidence, the sideward looking irradiance is proportional to the scalar irradiance, a very useful quantity for biology in the ice and ocean. This result is relied upon as a justification for the unusual orientation of the irradiance sensors on the string. While it may well be an accurate and useful result, I felt the presentation and examination of the modelling this result is based on is the weakest part of the paper and should be improved before final publication. If this tool becomes broadly used, this result will be widely used and cited, so it should be well founded here. Beyond that my comments below are mostly minor.
We included further discussion of the applicability of the main underlying assumption into the manuscript. Of course this equivalency can only be valid for light fields that are close to isotropic. Particularly for more directional light fields above the ice, in thin ice or just underneath the ice surface this equivalency does not hold up. Nevertheless we are convinced that sideward looking irradiance data can provide crucial insights also in these situations.

Regarding the modelling and interpretation of the model results:

1.  It appears there was no attempt to run the model for different wavelengths, given that the absorption coefficient is fixed. I feel it would be worth confirming that the factor-of-four difference to spherical irradiance holds at other wavelengths

Indeed, the light field modelling was only conducted in a monochromatic/broadband fashion, so it is not specific to any wavelength per se. Light transfer in sea ice is mostly governed by scattering, which is nearly wavelength independent, so we do not see a reason for spectrally resolved modelling.

2.  How were the absorption and scattering coefficients and asymmetry parameter(a/b/g in §2.5) chosen?   No references or reasons are given for the chosen values. Based on the estimates from Light et al (2008, 10.1029/2006JC003977), b=250 m-1seems low for the surface scattering layer (SSL), though b=25 m-1 is in their range for interior ice.  Warren et al (2006, 10.1364/AO.45.005320) give much lower values of a for pure ice.  Is a=0.15 based on measurements of sea ice?  Finally, g=0.9 is very low according to Light et al, who argued that the value they used, 0.94, "is probably too low for the majority of actual g values appropriate for sea ice."

Parameters were chosen to roughly fit the observed albedo and transmittance. As this modelling is aimed at the understanding of the relationship between scalar and sideward planar irradiance and not inverse retrieval of ice IOP, we do not try to exactly match parameters for a direct comparison of the chain data. We upgraded our statement in section and added literature references "These parameters were chosen guided by values previously used in the literature [Ehn et al., 2008b; Light et al., 2008; Petrich et al., 2012] and adjusted so that they resulted in calculated ice albedo and transmittance values very similar to our observations.."

Essential for this calculation is the effective scattering coefficient b' = b*(1-g) [e.g. Petrich et al 2012]. For numerical robustness and speed reasons it is common practice to run simulations at a lower g with appropriately lowered b.

3. The model has isotropic downwelling radiance above the ice, which is a reasonable approximation under thick clouds, which admittedly describes the high Arctic most of the summer. Still, it would be useful to see if and at what depths this relationship holds with a  direct  beam,  something  which  will  be  azimuthally dependent  and  potentially spectrally dependent.

This relationship (as any relation between scalar/planar irradiance and radiance) can of course only hold true for strongly diffuse light fields, such as present in the asymptotic regime or under overcast skies. We added the following statement to the Limitations section making this more clear: "However these equivalencies can of course only be valid for undirected diffuse and azimuthally homogenous light fields. While such diffuse light fields are prevalent in most in and under-ice scenarios, more directional light fields can occur during cloud-free conditions particularly above the ice surface, within the first layers and if a surface scattering layer is absent."

4. Scattering obviously dominates in the SSL, so it might not matter, but was the density used to account for the fact that the air in the SSL reduces the absorption since only40-50% of the path is ice?

As stated in section 2.5 we use a homogeneous plane parallel model and thus did not explicitly include air volume. As you noticed, Scattering is by far the dominant parameter, while the choice of ice absorption coefficient has very minor impact on the results.

5.  Showing some comparisons of the modelled normalized irradiance and extinction coefficient with the observed values would give readers more confidence in relying on the model result giving the proportionality between sideward and scalar irradiance.
This comparison can be easily made by the reader when comparing figures 5 and 9. The main idea behind our modelling was not a perfect parameter retrieval and data assimilation for that case, but a general model geometry roughly guided by our scenario. As we did not try to exactly reproduce measured results we do not present measurements and model in the same graph. Making the two fit in this one scenario is purely a matter of tweaking enough parameters and thus -from our viewpoint- does not provide any more scientific insight. To make this clear, we added the following statement in section 2.5 "The goal of this modelling analysis is the general evaluation of the sidewark-looking sensor orientation and not an exact reproduction of the deployment situation, thus we did not further tune the optical parameters to a perfect fit to the observations."

Other comments:

6.  Line 77:  Was the transparent heat shrink checked to make sure it was spectrallyneutral, or could it have altered the response curves shown in Figure 2?
We thank you very much for pointing out this issue. Following your suggestion, we performed additional experiments towards the characterization of the spectral transmittance of the heat shrink. Unfortunately heat shrink is a cheap industrial product, that is poorly controlled in terms of optical stability. We thus added the following statement to section 4.2: "Part of this uncertainty could also originate from the poorly characterised spectral transmittance of the heat shrink covering the sensors. Manufacturing differences and material aging make it difficult to precisely account for spectral transmittance of the heat shrink. A lab experiment revealed highest heat shrink transmittance for the blue channel, with 3% relative reduction in the red, 6% in the green and 21% in the clear channel."

7. Line 79: 48 sensors spaced 5 cm apart gives a total chain length (from first sensor to the last) of 235 cm.
Corrected accordingly

8. Line 95: Presumably one of the 'eight's should be 'six'.
Thank you for catching this. The sections are six sensors long. We corrected this accordingly.

9. Line 155: I would write 'channel' after 'clear' since 'in the clear' is an expression.
Corrected accordingly

10. While the boundary effects at the bottom of the ice are explained in section 3.1, the increase in irradiance between 0.2 and 0.3 m, just below the SSL, is not mentioned. It should at least be pointed out and possible reasons for it discussed.  Is it an effect of the hole, or are the lower sensors seeing more of a nearby bare area or pond? It could also be a real effect of the boundary. The light will refract on entering the ice from the air in the SSL, meaning it will wind up being more downward directed just below the surface, with very little light travelling nearly horizontally (since that would be reflected at the ice surface), so perhaps a side view a few cm below the air(SSL)-ice boundary is actually darker than after some scattering in the ice. I'm not

sure you can come with a definitive answer for why the measured radiance increases in that layer, but it shouldn't be ignored.

We thank you for pointing out this gap in our explanations. We are uncertain what exactly causes this effect, so we added the following statement including your suggestions: "Another notable feature of the profile is an increase between 0.2 and 0.3m. It could result from locally enhanced scattering, an effect of the sampling hole, an effect of the adjacent pond or refraction at the lower boundary of the scattering layer or the water-line."

11. Line 195: Figure 5 doesn't show any data after the 14 September snowfall, so it is illustrating exponential decay within the SSL.

We rephrased this paragraph to be more accurate.

12. The paragraph starting at line 215 should address why the clear channel has lower bulk extinction than any of the others, rather than being between the R and the G/B channels.

We thank you for pointing out this mystery! This effect does not appear during our recent spring 2020 deployments and is much milder during the fall 2020 deployment. It could thus be related either to melt pond vicinity or instrumental issues. To highlight that, we added the following statement: "It remains unclear to us why this prototype shows lower attenuation in the clear channel than the green and blue channels, instead of the expected values between the red and the blue and green channels. This effect did not occur in the deployments during spring 2020 (not described here), and thus seems to either be related to instrument uncertainties or the influence of close by melt ponds."

13. I would re-write line 225-226 to read 'with attenuation coefficients for the clear channel rising from 0.8 m-1 to 1.0 m-1 and for the red from 1.1 m-1 to 1.3 m-1.'

Corrected accordingly

14. Figure 3 is not referenced in the text.

We added the missing reference to the description of the deployment.

15. Figure 4: I would specify in the caption that it is the natural logarithm of sensor count that is plotted.

We added the word 'natural'.

16. I think Figure 6 would be more readable with a logarithmic scale on the color axis,or separate color bars put together to allow seeing the variation within the ice. The inclusion of Figure 7 allows for seeing details at the surface.

We respectfully disagree. Purpose of figure 6 is an overview about the values encountered throughout the ice column. Artificially blowing up small differences within the ice would distort the perception of relative importance of different layers. While differences inside the ice are interesting, they are also far less significant.

17. Since the focus of Figure 11 is the colour of the light, it might work better to use a constant brightness. When I look at it, on screen or paper, the colour information is largely drowned out by the brightness variations.

As both reviewers did not seem to understand this figure well and we did not manage to generate a clearer representation, we deleted this figure and rephrase the respective section accordingly: "The four spectral bands of the light sensor chain also

allow a simple assessment of light color and spectral changes over time. Our first results suggest that there is potential to detect at least transient high concentrations of in-ice algae by this light sensor chain, either in RGB plots or simple band ratios similar to remote sensing algorithms."